# Diagnostic Performance of Ag-RDTs and NAAT for SARS-CoV2 Identification in Symptomatic Patients in Catalonia

**DOI:** 10.3390/v13050908

**Published:** 2021-05-14

**Authors:** Luca Basile, Víctor Guadalupe-Fernández, Manuel Valdivia Guijarro, Ana Martinez Mateo, Pilar Ciruela Navas, Jacobo Mendioroz Peña

**Affiliations:** 1Sub-Directorate General of Surveillance and Response to Public Health Emergencies, Public Health Agency of Catalonia, Generalitat of Catalonia, 08005 Barcelona, Spain; vguadalupe@gencat.cat (V.G.-F.); manuel.valdivia@gencat.cat (M.V.G.); a.martinez@gencat.cat (A.M.M.); pilar.ciruela@gencat.cat (P.C.N.); jmendioroz@gencat.cat (J.M.P.); 2Unitat de Suport a la Recerca de Catalunya Central, Fundació Institut Universitari per a la Recerca a l’Atenció Primària de Salut Jordi Gol i Gurina, 08272 Sant Fruitós de Bages, Spain; 3CIBER Epidemiologia y Salud Pública (CIBERESP), Instituto Salud Carlos III, 28029 Madrid, Spain

**Keywords:** SARS-CoV-2 infection, antigen test, surveillance, pre-symptomatic, symptomatic, positivity rate

## Abstract

The use of rapid antigenic tests (Ag-RDTs) to diagnose a SARS-CoV-2 infection has become a common practice recently. This study aimed to evaluate performance of Abbott Panbio^TM^ Ag-RDTs with regard to nucleic acid amplification testing (NAAT) in the early stages of the disease. A cohort of 149,026 infected symptomatic patients, reported in Catalonia from November 2020 to January 2021, was selected. The positivity rates of the two tests were compared with respect to the dates of symptom onset. Ag-RDTs presented positivity rates of 84% in the transmission phases of the disease and 31% in the pre-symptomatic period, compared to 93% and 91%, respectively, for NAAT. The detection of many false negatives with Ag-RDTs during the pre-symptomatic period demonstrates the risk of virus dissemination with this diagnostic technique if used outside the symptomatic period.

## 1. Introduction

Currently, due to its high sensitivity and specificity, nucleic acid amplification testing (NAAT) has become the gold standard for the detection of severe acute respiratory syndrome coronavirus 2 (SARS-CoV-2) [1]. Ag-RDTs enable sensitive detection of high viral loads (Ct values ≤25 or >10^6^ genomic virus copies/mL) which usually arise in the pre-symptomatic (1–3 days before symptom onset) and early symptomatic phases of infection (within the first 5–7 days of infection) [2,3].

Although sensitivity is lower, especially in asymptomatic patients, Ag-RDTs offer advantages in terms of speed and low-cost strategies compared to NAAT [4,5].

In December 2020, the European Union adopted a common response for the use, validation and recognition of Ag-RDTs as a diagnostic method to detect SARS-CoV-2 infection [6]. The European Centre for Disease and Control (ECDC) agreed on a selected list of validated Ag-RDTs along with recommendations on their use for the diagnosis of COVID-19, based on the best available evidence to date, from systematic research of clinical trials [4]. Panbio^TM^ COVID-19 Ag Rapid Test Device (Abbott Diagnostics, Jena, Germany) started to be used massively in Catalonian primary care centers and hospitals at the end of October.

Although indications for the use of Ag-RDTs were limited, a mitigation strategy was implemented in order to relieve pressure on overburdened laboratories [7]. To overcome the bottleneck of NAAT testing, the use of Abbott Panbio^TM^ Ag-RDT was widespread for early detection of positive cases at the community level, especially for contact tracing, routine COVID-19 surveillance in long-term residential care settings and mass testing [8].

## 2. Aim

To evaluate the diagnostic performance of SARS-CoV-2 NAAT and Abbott Panbio^TM^ Ag-RDT in pre-symptomatic and symptomatic phases of COVID-19 patients on the basis of epidemiological strata.

## 3. Materials and Methods

An observational retrospective study was conducted on a cohort of symptomatic patients diagnosed with COVID-19 in Catalonia, from 1 November 2020 to 31 January 2021. A confirmed case of SARS-CoV-2 was defined as a patient with at least one, NAAT or Ag-RDTs or ELISA IgM, positive result. Nasopharyngeal and blood samples were collected by trained nurses. For NAAT, reverse-transcriptase polymerase chain reaction testing was performed within 24 hours of specimen collection, targeting SARS-CoV-2 E or N and RdRP or genes S and N [9]. Although several Ag-RDTs were available according to the ECDC recommendations, the Abbott Panbio^TM^ Ag-RDT was selected after several validations as the only test kit to be used in the Catalan public health system [4].

Data were analyzed from the Epidemiological Repository of Catalonia (REC), an electronic registry used by the Catalonian Epidemiological Surveillance Network (XVEC) which automatically receives results of Ag-RDTs, NAAT and serological tests from National Health Service laboratories, and also from several private entities.

Epidemiological (age, date of diagnosis, date of symptoms onset), exposure (health care workers, nursing home residents), outcome (hospitalization, exitus) and setting (higher incidence rate/lower incidence rate period) characteristics for selected patients were analyzed. Assessment of onset of symptoms was mainly based on retrospective self-reports by patients after diagnosis confirmation.

For each patient included, every NAAT and/or Ag-RDT test performed at any time before and after diagnosis was considered. The difference between date of collection (for NAAT) or date of result (for Ag-RDTs) and date of symptom onset was calculated, and tests performed from 14 days before to 14 days after symptom onset were selected. Results are presented as positivity rates. A two proportion Z-test was carried out to determinate P-values and confidence intervals of the positivity rate’s differences between Ag-RDTs and NAAT. The Epidat program (V.4.2, Conselleria de Sanidade, Xunta de Galicia) was used for the statistical tests.

## 4. Results

Between 1 November 2020 and 31 January 2021, 219,138 patients received a confirmed diagnosis of COVID-19. Among them, 149,026 were symptomatic (68%). Demographic and epidemiological characteristics are shown in Table 1.

During the studied period, 7-day cumulative incidence rates (cases per 100,000 inhabitants) varied between 101 and 350, being above 150 for 9 weeks (from 1 November to 15 November and 14 December to 31 January) and below 150 for 4 weeks (from 16 November to 13 December).

A total of 139,462 Ag-RDTs and 218,724 NAAT were performed on symptomatic patients. Each patient took an average of 2.4 tests (SD 2.27): 0.9 Ag-RDTs (SD 0.78) and 1.5 NAAT (SD 2.19).

Positivity rates were 80% and over between +1 and +7 days from the symptom onset for Ag-RDTs and between −2 and +10 days for NAAT. The highest probability of detecting a positive case was on the third day after symptom onset (93.0% of positive tests) for Ag-RDTs and on the first day after symptom onset for NAAT (95.5%) (Table 2).

During the transmission period, from −2 to + 7 days from symptom onset [10], the overall positivity rate was 84.2% for Ag-RDTs and 93.1% for NAAT (*p*-value < 0.001) (Table 3).

The highest difference in positivity rate between the two tests was found in the pre-symptomatic period, the second day before symptom onset being the day with the highest difference (Figure 1; Table 2 and Table 3). During the two days before symptom onset, the positivity rate was 30.8% for Ag-RDTs and 90.9% for NAAT (*p*-value < 0.001). Differences between the two tests were higher in the 15–44 age group (28% for Ag-RDTs vs. 91% for NAAT), and lower in the 0–14 age group (40% for Ag-RDTs vs. 95% for NAAT) and in patients with severe outcomes (36% vs. 83% in hospitalized patients and 39% vs. 82% in exitus patients) than in patients with milder symptoms (Table 3, Appendix A).

The best performance of Ag-RDTs was observed in 5–14 age group, in pre-symptomatic (42.6%), symptomatic (90.8%) and transmission periods (87.4%) (Table 3).

In health workers and residents of nursing homes the use of Ag-RDTs was limited with respect to NAAT (Appendix A), and positivity rates for Ag-RDTs were lower than in the rest of patients during the transmission period (79.9% in health care workers and 74.9% in residents of nursing home) and especially during the pre-symptomatic period (22.3% in health care workers and 24.4% in nursing home residents) (Table 3).

In higher incidence conditions, greater performances by both Ag-RDTs and NAAT were observed, especially in Ag-RDTs during pre-symptomatic period and post-transmission period (Table 3).

From the eighth day after symptom onset, the positivity rate of Ag-RDTs began to drop below 80% and from the 12th day to drop below 30%, whereas in NAAT it remained higher than 70% until the 14th day (Figure 1). Differences in drop rate were observed after the 12th day in all categories for those under 65 years old and in health workers; for those over 65 who were hospitalized or died, the Ag-RDT rate remained above 50% until the end of the observed period (Table 2, Appendix A).

## 5. Discussion

According to current guidelines on the use of Ag-RDTs [11], authorized Ag-RDTs, and especially Abbott Panbio^TM^, have reported consistently high levels of specificity, but they do not achieve equal levels of sensitivity compared to NAAT [12,13].

Due to the evidence of SARS-CoV-2 transmission in pre-symptomatic and asymptomatic patients [14,15], results presented in this study support the statement regarding a notably higher reliability of NAAT compared to Abbott Panbio^TM^ Ag-RDTs during pre-symptomatic phases. Moreover, the high number of false negative cases detected in our study with Ag-RDTs during the pre-symptomatic period indicates that in suspected patients with no symptoms the negative antigen result should be confirmed with a NAAT [16]. Additionally, results from molecular assays during post-symptomatic period should take into account the possible presence of a residual SARS-CoV-2 RNA load (quantification cycle value greater than 30) with a subsequent negative Ag-RDT result, which could be pointed as a confounder of “real positive” cases with ongoing virus replication [17,18].

Our results showed an accurate performance of Ag-RDTs in the age-group 5–14, unlike other studies [19,20]. The differences between age-groups could be related to differences in SARS-CoV-2 viral load [20,21]. However, age-related differences in the performance of Ag-RDT remain unknown. These results highlight the need to assess whether the widespread use of Ag-RDTs in pediatric patients can help to prevent and control the covid-19 pandemic. For this reason, more research in the clinical assessment of Ag-RDTs being used on patients of extreme age is needed.

Furthermore, we found greater performance of Ag-RDTs in a higher prevalence setting, especially from 8th day after symptom onset, probably due to higher viral load associated with higher virus dissemination [22].

In this sense, it should also be noted that the higher Ag-RDTs performance from the 7th day after symptom onset in adults aged 65 and over and especially in hospitalized patients and the deceased could be explained by the higher duration of infectious viral shedding in those groups. Previously published studies indicated that the presence of a high viral load of SARS-CoV-2 on admission was associated with a higher risk of respiratory failure and mortality [23,24]. It is, therefore, necessary to take into account that the duration of infectious viral shedding currently described is longer for severely-ill patients, which explains the higher positivity rates of NAATs and RDTs over time in this group [25].

Nevertheless, this study presents limitations: Assumptions about date of symptom onset may be affected by recall bias and uncertainty due to inaccurate recall of the symptoms onset by the study subjects [26]. Comparison of the cycle threshold (Ct) values of RT-PCR was not possible due to the lack of data availability in the existing databases. Furthermore in this respect, the high variability between analytical interpretive methods and laboratory factors must be taken into account [27]. Moreover, results should be taken with caution as the present research does not apply a single-subject design and the two techniques were not replicated in parallel for all patients. For this reason, this study was not intended to provide estimates of sensitivity and specificity, but to evaluate the appropriateness of the use of both testing procedures in a pandemic containment strategy.

The use of Ag-RDTs in pre-symptomatic close contacts of positive SARS-CoV-2 cases is a common practice in Catalonia. In the last few months, XVEC has observed some relaxation in the compliance of quarantine measures, usually by individuals who obtained negative Ag-RDT during the asymptomatic phase (and who are likely to end up with a positive NAAT). This misbehavior, in addition to the possible high number of Ag-RDT false negatives, leads us to think that the use of this type of test could even be counterproductive in certain scenarios, leading to a possible increase in the spread of the virus in the community.

Appropriate use of Ag-RDTs included scenarios with a high expected prevalence of disease, when NAAT capacities were not available and receiving timely results is critical — for example, for contact tracing purposes [4]. In this sense, the studied period presented a 7-day incidence rate of over 100 cases per 100,000 inhabitants and a NAAT positivity rate of over 5%. The difficult situation in Catalonia in that period and the saturation suffered by the laboratories justifies the choice of including Ag-RDTs in the containment strategy by public health authorities.

Findings from this paper do not aim to support nor to retract Ag-RDTs suitability, but to limit its use to the largest possible extent, by only and exclusively setting its performance as a response to scenarios that are clearly recommended by the scientific community and Public Health Agencies.

## Figures and Tables

**Figure 1 viruses-13-00908-f001:**
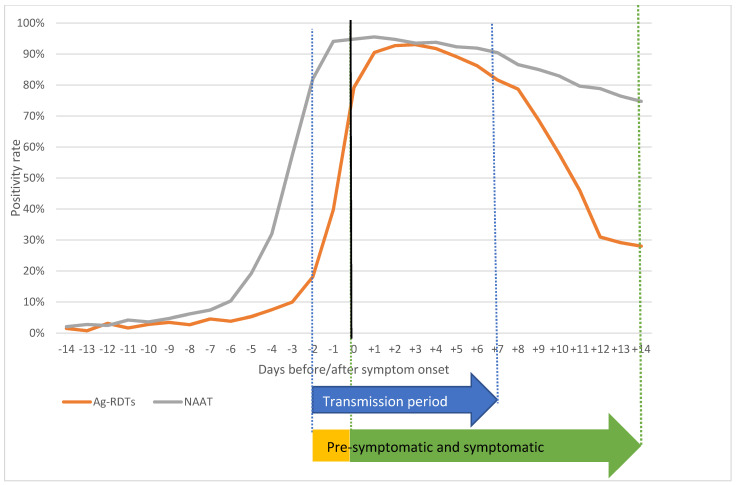
Positivity rates of Ag-RTDs and NAAT with respect to symptom onset. Ag-RDTs: rapid antigenic tests; NAAT: nucleic acid amplification testing.

**Table 1 viruses-13-00908-t001:** Socio-demographic and epidemiological characteristics of patients with confirmed COVID-19.

	Total		Symptomatic		Asymptomatic		*p*-Value
	*N*	%	*N*	%	*N*	%	
Diagnosed patients	219,138		149,026	68.0%	70,112	32.0%	
Sex							
Female	117,220	53.5%	81,034	54.4%	36,186	51.6%	<0.001
Male	101,918	46.5%	67,992	45.6%	33,926	48.4%	<0.001
Age groups							
0–14	28,225	12.9%	12,723	8.5%	15,502	22.1%	<0.001
15–44	87,108	39.8%	63,977	42.9%	23,131	33.0%	<0.001
45–64	63,103	28.8%	47,437	31.8%	15,666	22.3%	<0.001
65–80	24,831	11.3%	16,796	11.3%	8035	11.5%	0.194
>80	15,871	7.2%	8093	5.4%	7778	11.1%	<0.001
Residents of nursing home	7977	3.6%	2186	1.5%	5791	8.3%	<0.001
Health care workers	4539	2.1%	3682	2.5%	857	1.2%	<0.001
Clinical evolution							
Hospitalized	8361	3.8%	7021	4.7%	1340	1.9%	<0.001
Exitus	4109	1.9%	2313	1.6%	1796	2.6%	<0.001
Diagnostic tests							
Ag-RDTs	97,576	44.5%	78,560	52.7%	19,016	27.1%	
With any negative NAAT			5796	7.4%			
NAAT	106,484	48.6%	58,667	39.4%	47,817	68.2%	
With any negative Ag-RDTs			14,522	24.8%			
Ag-RDTs + NAAT	14,561	6.6%	11,741	7.9%	2820	4.0%	
ELISA IgM	517	0.2%	58	0.0%	459	0.7%	
Total tests done							
Ag-RDTs *	181,276	32.60%	139,462	38.90%	41,814	21.10%	
Before SO * (Positives)			31,063 (3212)	22.3% (10.3%)			
After SO * (Positives)			108,399 (88,743)	77.7% (81.9%)			
NAAT *	375,388	67.40%	218,724	61.10%	156,664	78.90%	
Before SO * (Positives)			127,517 (21,568)	58.3% (16.9%)			
After SO * (Positives)			91,207 (67,722)	41.7% (74.3%)			

* Ag-RDTs: rapid antigenic tests; NAAT: nucleic acid amplification testing; SO: symptoms onset.

**Table 2 viruses-13-00908-t002:** Total tests, positive tests, positivity rates and differences between Ag-RDTs and NAAT with respect to symptom onset.

Days	Ag-RDTs			NAAT			▲%	CI−	CI+	*p* Value
	Tests	Positives	%	Tests	Positives	%				
−14	396	6	1.5%	1100	23	2.1%	−0.6%	−2.2%	1.1%	0.617
−13	384	3	0.8%	1072	30	2.8%	−2.0%	−3.5%	−0.5%	0.038
−12	445	14	3.1%	1132	28	2.5%	0.7%	−1.3%	2.7%	0.567
−11	538	9	1.7%	1352	57	4.2%	−2.5%	−4.2%	−0.9%	0.010
−10	602	17	2.8%	1272	46	3.6%	−0.8%	−2.6%	1.0%	0.452
−9	666	23	3.5%	1420	67	4.7%	−1.3%	−3.1%	0.6%	0.226
−8	852	23	2.7%	1461	90	6.2%	−3.5%	−5.2%	−1.7%	0.000
−7	1163	53	4.6%	1729	128	7.4%	−2.8%	−4.6%	−1.1%	0.003
−6	1326	51	3.8%	1688	174	10.3%	−6.5%	−8.3%	−4.6%	0.000
−5	1624	86	5.3%	1962	376	19.2%	−13.9%	−16.0%	−11.8%	0.000
−4	1884	141	7.5%	2249	718	31.9%	−24.4%	−26.8%	−22.1%	0.000
−3	2538	253	10.0%	3162	1814	57.4%	−47.4%	−49.5%	−45.3%	0.000
−2	3266	592	18.1%	5167	4235	82.0%	−63.8%	−65.5%	−62.1%	0.000
−1	4647	1849	39.8%	14,360	13,512	94.1%	−54.3%	−55.8%	−52.8%	0.000
Symptom onset	20,806	16,476	79.2%	12,190	11,559	94.8%	−15.6%	−16.3%	−15.0%	0.000
+1	26,770	24,232	90.5%	8800	8406	95.5%	−5.0%	−5.6%	−4.4%	0.000
+2	20,285	18,806	92.7%	6267	5937	94.7%	−2.0%	−2.7%	−1.4%	0.000
+3	12,731	11,845	93.0%	4555	4260	93.5%	−0.5%	−1.3%	0.4%	0.257
+4	7223	6629	91.8%	3388	3178	93.8%	−2.0%	−3.1%	−1.0%	0.000
+5	4024	3587	89.1%	2673	2468	92.3%	−3.2%	−4.6%	−1.8%	0.000
+6	2410	2078	86.2%	2555	2348	91.9%	−5.7%	−7.5%	−3.9%	0.000
+7	1876	1531	81.6%	2227	2014	90.4%	−8.8%	−11.0%	−6.6%	0.000
+8	962	757	78.7%	2148	1860	86.6%	−7.9%	−10.9%	−4.9%	0.000
+9	728	500	68.7%	2844	2417	85.0%	−16.3%	−20.0%	−12.6%	0.000
+10	656	379	57.8%	2989	2479	82.9%	−25.2%	−29.3%	−21.1%	0.000
+11	554	255	46.0%	2816	2243	79.7%	−33.6%	−38.1%	−29.1%	0.000
+12	507	157	31.0%	2434	1919	78.8%	−47.9%	−52.3%	−43.4%	0.000
+13	436	127	29.1%	2490	1903	76.4%	−47.3%	−52.0%	−42.6%	0.000
+14	414	116	28.0%	2141	1600	74.7%	−46.7%	−51.6%	−41.9%	0.000

Ag-RDTs: rapid antigenic tests; NAAT: nucleic acid amplification testing; ▲: difference; CI: Confidence interval.

**Table 3 viruses-13-00908-t003:** Positivity rates of Ag-RTDs and NAAT aggregated by illness/transmission phases for epidemiological strata.

	Pre-Symptomatic Period (−2; −1)			Symptomatic Period (0; +14)			Transmission Period (−2; +7)			Post-Transmission Period (+8; +14)		
	Ag-RDTs	NAAT	▲%	Ag-RDTs	NAAT	▲%	Ag-RDTs	NAAT	▲%	Ag-RDTs	NAAT	▲%
Total	30.8%	90.9%	−60.0%	87.1%	90.2%	−3.1%	84.2%	93.1%	−8.9%	53.8%	80.7%	−26.9%
0–4	32.1%	95.8%	−63.7%	90.0%	93.5%	−3.5%	84.3%	94.7%	−10.4%	59.0%	84.3%	−25.3%
5–14	42.6%	95.3%	−52.7%	90.8%	94.7%	−3.9%	87.4%	95.7%	−8.3%	49.2%	80.6%	−31.5%
15–44	28.1%	90.5%	−62.3%	86.7%	89.6%	−2.8%	83.8%	93.4%	−9.6%	43.8%	79.3%	−35.5%
45–64	29.4%	90.1%	−60.7%	86.7%	90.2%	−3.5%	83.9%	93.2%	−9.3%	53.5%	81.0%	−27.6%
65–80	37.0%	91.3%	−54.2%	87.8%	92.2%	−4.4%	85.3%	93.5%	−8.1%	70.6%	85.3%	−14.7%
+80	37.1%	88.5%	−51.4%	87.8%	87.0%	0.9%	84.3%	88.3%	−4.0%	77.1%	83.4%	−6.3%
Health workers	22.3%	86.1%	−63.8%	81.1%	82.6%	−1.5%	79.9%	89.7%	−9.9%	29.5%	76.7%	−47.1%
Nursing home residents	24.4%	84.9%	−60.5%	79.7%	82.0%	−2.3%	74.9%	84.7%	−9.8%	50.8%	77.4%	−26.6%
Hospitalized	35.5%	82.5%	−47.1%	78.7%	85.4%	−6.7%	76.2%	85.4%	−9.3%	69.6%	82.2%	−12.6%
Exitus	38.8%	81.8%	−43.0%	87.3%	85.0%	2.3%	83.4%	84.2%	−0.8%	82.4%	82.8%	−0.5%
Higher incidence rate *	31.3%	91.0%	−59.7%	87.4%	90.3%	−3.0%	84.4%	93.2%	−8.8%	54.5%	80.7%	−26.2%
Lower incidence rate *	28.1%	90.3%	−62.1%	85.8%	89.6%	−3.8%	83.0%	92.6%	−9.6%	49.9%	80.9%	−31.0%

Ag-RDTs: rapid antigenic tests; NAAT: nucleic acid amplification testing. * Cut-off point fixed within 7-day incidence rate of 150 cases per 100,000 inhabitants.

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
