# Peer review of "Diagnostic Performance of Ag-RDTs and NAAT for SARS-CoV2 Identification in Symptomatic Patients in Catalonia"

_viruses, 2021, doi:10.3390/v13050908_

Round 1

Reviewer 1 Report

This manuscript aims to evaluate the performance of SARS-CoV-2 NAAT and Ag-RDTs (Abbott Panbio) in pre-symptomatic and symptomatic phases of COVID-19 infection by performing a retrospective study on a cohort of symptomatic patients diagnosed with COVID-19 in Catalonia, Spain, from November 1, 2020 to January 31, 2021.  

Overall, the manuscript analysis a substantial amount of data, provides important information, it’s actual and can deserve published. However, there are several points that need clarification and/or could be improved by further analysis and discussion.

Having the purpose to evaluate the performance of SARS-CoV-2 NAAT and Ag-RDTs, it should be more clearly stated throughout the manuscript that the only Ag-RDTs in evaluation is Abbott Panbio test, that is only referred in section 3 (Materials and Methods), and even so, without full detail reference, that should be provided. I believe it should be Panbio™ COVID-19 Ag Rapid Test Device (Abbott Diagnostics, Jena, Germany). The sole use of this Ag-RDT should be clearly stated throughout the manuscript, starting from the Abstract.

On the other hand, the NAAT protocols used are not described. Although NAAT is recognized as the gold standard for SARS-CoV-2 diagnosis and there is a global scientific agreement that should be the first choice for symptomatic patients, there are several protocols available with different performances, namely in sensitivity, specificity and detection of SARS-CoV-2 variants. A full evaluation of this study is only possible with access to this missing information.

Moreover, although the authors state that “this study was not intended to provide estimates of sensitivity and specificity, but to evaluate the appropriateness of the use of both testing procedures in a pandemic containment strategy.” (lines 148-150), the evaluation of the subset of patient’s data that had results for both tests is essential in this study. An integrate analysis would enable the detection of disagreements (namely, % of only positive NAAT tests, % of only positive Panbio Ag-RDT, …), comparison of detected Cts/genomic virus copies/ml versus positive/negative Panbio Ag-RDT, etc. This would provide further insights in the evaluation of the appropriateness of both tests in containment strategies, since it is also commonly accepted that NAAT tests generally provide positive results several days after transmission/infectious period.

The discussion of the study limitations is also missing. One especially important in this study is related to the clarification of how was determined the symptom onset: clinical report or patient recall after diagnosis confirmation?

Specifically, I have some other comments and suggest the following:

  • In the text, all reference numbers in square brackets [ ] should be placed before the punctuation, and are always after punctuation. Lines 31, 34, 36, 39, 42, 45, 48, 59, 128, 134, 136, 137, 142, 145, 160.
  • Lines 58-59; please correct: “…the ECDC recommendations, the Panbio™ COVID-19 Ag Rapid Test Device (Abbott Diagnostics, Jena, Germany) is was selected after several validations as the only test kit to be used in the Catalan public health system.”
  • Lines 70-71, please correct: “Results were are presented as positivity rates.”
  • Table 1: please add commas to all numbers larger than 999.
  • Lines 94-95: different letter format, please correct.
  • Figure 1. Please add reference as adapted from X, since the definition of transmission period was defined by previous reports.
  • Lines 175-181: different format, please correct.
  • References are not in agreement to Viruses Instructions and should be corrected.
  1. Manuscripts should follow:

Author 1, A.B.; Author 2, C.D. Title of the article. Abbreviated Journal Name YearVolume, page range; DOI.

  1. References of Websites and online resources miss the information of acession, namely for references 4, 6, 8 and 10, stated as “Available online: http://... (accessed on date).”

Full instructions for authors can be accessed at:  https://www.mdpi.com/journal/viruses/instructions

Reviewer 2 Report

The manuscript by Basile et al. evaluates the performance of a rapid antigenic diagnostic test compared to nucleic acid amplification detection test for SARS-CoV2 in a total of 149,026 COVID-19 diagnosed patients in Catalonia, Spain.

The study lacks scientific novelty related to SARS-CoV2 detection, as the results are the expected ones for the test used, according to the European Center for disease prevention and control published recommendations about rapid antigenic tests. However, taking into account the extraordinary importance and impact of this disease worldwide, in my opinion, a big set of data as the one presented in this study is still valuable for confirmation and validation of rapid antigenic tests indications in certain pandemic condition scenarios.  

Nevertheless, the manuscript should be improved in several aspects in order to be accepted for publication in Viruses.

Main points:

1.- Authors should remove the word “accuracy” from the title of the manuscript. Diagnostic accuracy of a technique is its ability to detect the pathogen when it is present and give a negative result when the pathogen is not present. To evaluate accuracy of a diagnostic method, diagnostic parameters such us sensitivity, specificity, positive and negative predictive values, likelihood ratios and post-test probability have to be evaluated. As this is not the case of this study, I strongly recommend removing the word “accuracy”. I suggest “performance” instead.

2.- The study presented in the manuscript has been performed in a time period covering the descent of the second wave and the rise of the third wave, including its peak. This means that data correspond to both, a high prevalence setting and a low prevalence setting.

As the diagnostic parameters depend on the prevalence of the pathogen, it would be very interesting to discriminate these two situations. The rapid antigenic test used is expected to perform very differently in a high prevalence condition (in which it is expected to have a high PPV) than in a low prevalence condition (in which it is expected to have a high NPV). I suggest the authors to conduct the data analysis in these two separate conditions and evaluate the positivity rate of rapid antigenic tests compared to nucleic amplification test.

3.- I see a putative scientific novelty of this study in the performance of the Ag-RDT used in the age group 0-14. Authors state an accurate performance compared to previous studies (page 7, lines 135-137), although they do not go deeper in this issue. I think it is very interesting, specially taking into account that there are no current vaccination programs addressed to this population group. I encourage authors to deeper analyze and to give more details on this issue.

4.- Figure 2 is not very informative, I would recommend removing it.

5.- Authors should describe NAAT method/s used.

Minor points:

1.- page 2, line 70: the sentence is not clear, rewrite it as follows, “tests performed from 14 days before to 14 days after symptom onset”.

2.- page 2, line 77. Rewrite the sentence as follows, “Among them 149,026 were symptomatic (68%).

3.- page 4, line 94-95, letter type seems to be different.

4.- page 4, line 94, remove brackets.

5.- page 7, lines 138-145, the positivity rate cannot indicate a higher duration of infectious virus shedding, the positivity rate can be explained by or according to a higher duration of virus shedding. Please revise this paragraph.

6.- page 7, line 154, rewrite as “(and who are likely to end up with a positive NAAT)”.

Round 2

Reviewer 1 Report

Most of my comments were addressed. However, I still think that some points could be improved before publication: 

    • Table 3 - Please correct last column title to "Post-transmission period" instead of "post-trasmission period"
    • Discussion can be improved by the addition of two points:  (1)Lines 146-148, the sentence presents a fact that is known for both techniques (Ag-RDTs and NAAT), only a “truly bad NAAT” would present lower sensitivity compared to an Ag-RDT test. The main point should be if using authorized Ag-RDTs would be efficient to prevent SARS-CoV-2 transmission, since it is also acknowledged that NAAT provide positive results during Post-transmission period. Since no Ct values are available and so, since this is not analysed in this work and symptom onset is determined by patient recall, some of the negative Ag-RDTs tests that had positive NAAT results could be in post-transmission period. There are works reporting the presence of symptoms in post-transmission period. This should be clearly stated.

(2)Lines 156-158, presents one of the most (if not the most) interesting result of this work: “an accurate performance of Ag-RDTs in the age-group 5-14”. Although correctly stated that “More research in the clinical assessment of Ag-RDTs in extreme ages is needed.”, this result could be positively highlighted regarding the use of Ag-RDTs to prevent SARS-CoV-2 transmission in this age group.

Reviewer 2 Report

Authors have fully addressed my comments. Therefore I recommend publication in Viruses. I just have some minor modifications to be made in the tables:

  • Table 3 has a different format than tables 1 and 2. Authors should check table format for all tables (I think horizontal lines are not allowed) and use for table 3 the same format as tables 1 and 2.
  • Column headings in table 3: change "TAR" to "Ag-RDT"; change "PCR" to "NAAT".
  • Asterisk at the last two lanes in table 3 should be explained at the table footnote. 
